# The Implementation of Preventive Health Measures in Small- and Medium-Sized Enterprises—A Combined Quantitative/Qualitative Study of Its Determinants from the Perspective of Enterprise Representatives

**DOI:** 10.3390/ijerph19073904

**Published:** 2022-03-25

**Authors:** Friederike E. Benning, Sandra H. van Oostrom, Femke van Nassau, Rosanne Schaap, Johannes R. Anema, Karin I. Proper

**Affiliations:** 1Center for Nutrition, Prevention and Health Services, National Institute for Public Health and the Environment, 3721 MA Bilthoven, The Netherlands; friederike.benning@rivm.nl (F.E.B.); sandra.van.oostrom@rivm.nl (S.H.v.O.); 2Department of Public and Occupational Health, Amsterdam Public Health Research Institute, Amsterdam University Medical Center, Vrije Universiteit Amsterdam, 1081 HV Amsterdam, The Netherlands; f.vannassau@amsterdamumc.nl (F.v.N.); r.schaap@amsterdamumc.nl (R.S.); h.anema@amsterdamumc.nl (J.R.A.)

**Keywords:** implementation, determinants, preventive health measures, small and medium-sized enterprise, exploratory study

## Abstract

The workplace is an ideal environment for promoting workers’ health. Nevertheless, preventive health measures are insufficiently implemented, especially in small and medium-sized enterprises (SMEs) with up to 250 employees. The aim of this study was to investigate determinants for the implementation of measures to prevent musculoskeletal and mental health disorders from the perspective of enterprise representatives in Dutch SMEs. An online survey was completed by 79 SME representatives (e.g., owners, HR professionals and occupational health and safety officers) in the cleaning, care, construction and transport sectors. In addition, semi-structured interviews were conducted with 18 enterprise representatives. The interview transcripts were analyzed using an inductive approach. Survey data showed that the focus of prevention efforts by SMEs is on improving working conditions and complying with legally required occupational health requirements, while lifestyle measures are rarely implemented. The determinants of implementation according to enterprise representatives were associated with 10 distinct themes. These were (1) available resources (both finances and staff), (2) complexity of implementation of measures, (3) awareness, (4) knowledge and expertise, (5) availability of time, (6) employer and worker commitment, (7) workers’ openness for measures, (8) communication, (9) workers’ trust and autonomy and (10) integration in organizational policy. These findings can serve as a support for developing strategies for implementing preventive health measures in SMEs.

## 1. Introduction

Musculoskeletal and mental health disorders are common among workers in the European Union [1]. In the latest European Working Conditions Survey [1], 44% of workers indicated that they suffered from back pain in the last 12 months, followed by muscular pain in the arms (42%), headache and eyestrain and overall fatigue (both 36%), muscular pains in the legs (30%) and anxiety (16%). These complaints not only affect the health and quality of life of individual workers, but they also burden enterprises and society as a whole through lost productivity due to incapacity to work, sickness-related absenteeism from paid and unpaid work and “presenteeism” (i.e., reduced productivity while at work) [2,3,4]. 

There are numerous preventive health measures available to target risk factors for musculoskeletal and mental health disorders at the workplace. These range from interventions targeting the work organization and the work environment to those targeting individual health-related behaviors [5,6]. However, their implementation lagsbehind. The implementation gap is particularly pronounced in small- and medium-sized enterprises (SMEs) with 250 or less workers [7,8,9,10]. For instance, the European survey of enterprises on new and emerging risks (ESENER) [10] showed that 54% of business with 250 or more workers had an action plan to prevent work-related stress. However, this was the case in only 40% of businesses with 50–249 workers and 31% of businesses with 10–49 workers. In addition, while 69% of businesses with 250 or more workers had used the service of an expert dealing with the ergonomic design and set-up of workplaces, only 56% and 40% of businesses with 50–249 workers and 10–49 workers, respectively, made use of these services. Thus, SMEs with up to 250 workers have the greatest need for an improved implementation of measures to prevent musculoskeletal and mental health disorders. 

A comprehensive literature review by Fan et al. (2020) on occupational health and safety research identified a scarcity of research in SMEs [7]. Only 17 out of 564 identified articles focused specifically on SMEs. The studies that have been conducted on SMEs suggest that it is difficult to manage worker health and safety due to factors like employers’ and workers’ lack of knowledge and competencies of the workplace risks and the enterprise’s occupational health and safety rights and legal obligations, low unionization rates and poor employment relations, as well as a lack of resources [7]. On the other hand, a recent research project by the European Agency for Safety and Health at Work in Micro and Small Enterprises suggests that drivers for taking up preventive efforts are the personal values of the owner/manager, the demands of workers, external drivers such as inspection and regulation and incentives like customers demanding a certification scheme as a prequalification [8]. 

To address the problem of poor implementation of preventive health measures targeting musculoskeletal and mental health disorders in Dutch SMEs, determinants of implementation in this specific context need to be identified and well understood. Therefore, the aim of this study was to provide an overview of the current use of preventive health measures to prevent musculoskeletal and mental health disorders in SMEs and to identify and examine the determinants of the implementation of those measures from the perspective of the employer or other enterprise representatives. The latter choice was made because the legal responsibility for ensuring safe and healthy working conditions in the Netherlands lies with the owner of a company or a person appointed by the owner. By gaining knowledge about determinants of the implementation, effective implementation strategies can be developed for protecting and improving the health of workers.

## 2. Materials and Methods

### 2.1. Study Design

A combined quantitative and qualitative study design was applied. Initially an online survey was carried out among SME employers (up to 250 workers) and other enterprise representatives in the cleaning, care, construction and transport sectors in the Netherlands. These sectors were chosen because of the high prevalence of musculoskeletal and mental health disorders among workers and relatively high related absenteeism [9]. Subsequently, semi-structured individual interviews were conducted with a selection of enterprise representatives who had filled in the survey. The aim of the interviews was to obtain more in-depth information.

### 2.2. Data Collection 

#### 2.2.1. Definition of Preventive Health Measures

In this research, a preventive health measure is defined as an attempt to improve musculoskeletal and mental health conditions in workplaces by means of targeted activities and initiatives. Such activities include, amongst others, changes in work organization and working conditions, engineering activities for the modification or installation of plant and equipment, training and behavioral changes. 

#### 2.2.2. Study Population Recruitment and Selection 

In order to reach enterprise representatives, three different strategies were applied. First, contact persons referred to us by existing contacts within sector organizations in the cleaning, care, transport and construction sectors were informed about the study and asked to distribute the survey among their SME members. In doing so, the contact persons were provided by the research team with a flyer with brief information about the study; a recruiting text to post on their website, social media or the newsletter and the link to the survey. Each contact person used their own existing and usual communication channels to approach employers. These included newsletters, social media sites and face-to-face contact. Second, several other organizations, such as local employer associations, the Netherlands Society of Occupational Medicine and a network for HR professionals, were approached and asked to distribute the recruitment material. Third, enterprise representatives participating in the survey were asked to send the survey to other enterprise representatives. The data collection took place from February 2021 until July 2021. In total, 79 respondents in our target group completed the survey. Although we could not specify the response rate because of this broad recruitment approach, we assume it was very low. 

For the semi-structured interviews, enterprise representatives across the cleaning, care, construction and transport sectors were recruited via the preceding survey. At the end of the survey, they could indicate whether they were willing to participate in an interview to obtain more in-depth information on the topic of the survey. We aimed for an equal number of enterprise representatives per industry and enterprise size. Eventually, all respondents who indicated to be willing to participate were interviewed. 

The Center for Clinical Expertise of the Dutch National Institute of Public Health and the Environment assessed the research proposal and classified the study as exempt from ethical review as it did not meet the criteria of the Medical Research Involving Human Subjects Acts. The center approved the study protocol. The study was performed in accordance with guidelines of good clinical practice and ethical principles as stated in the Declaration of Helsinki. Informed consent was obtained from all participants.

#### 2.2.3. Survey among Enterprise Representatives

The survey was divided into four parts. Part one inquired on the presence of the main risk factors for musculoskeletal disorders and mental health disorders in the enterprise derived from the Employer Monitor Working Conditions [11]. The second part asked whether specific preventive health measures had been implemented in the last 5 years (response categories: yes, no, measure already existed/do not know). A list of 6 and 11 preventive health measures targeting risk factors for musculoskeletal disorders and mental health disorders, respectively, were presented to respondents. Additionally, 6 general measures targeting worker health in general were presented. The above questions partly originated from the Dutch Employer Monitor Working Conditions, ESENER [10] and from the Dutch Employers Survey Working Conditions [9]. The survey also inquired to what extent employees were involved in deciding on which measures to implement (response categories: 6-point Likert scale from 1 = “to a very large extent” to 6 = “not, the owner decides”). In addition, respondents were asked to add measures that they used but that were not part of the provided lists. Part three comprised questions from the Measurement Instrument for Determinants of Innovations (MIDI) by Fleuren et al. [12] to map the determinants that affect the use of an innovation in practice into one of the following four categories: (1) characteristics of the innovation, (2) characteristics of the potential user of the innovation, (3) characteristics the organization where the potential user works and (4) characteristics of the socio-political context. For the purpose of this study, a total of 20 items were selected from this instrument in consultation with the research team. The instrument has an interval-scaled Likert response option (from 1 = “strongly disagree” to 5 = “strongly agree”). An opt-out option was also given. The fourth part contained 7 general questions aimed at collecting socio-demographic data of the enterprise and the person completing the survey. 

A small pilot of the survey was conducted within a group of employers (*n* = 3); representatives of sector organizations in the cleaning, care and transport sectors (*n* = 4); representatives from a national employer organization (*n* = 2) and one worker in the care sector in order to test its feasibility and comprehensibility. This led to minor adoptions.

#### 2.2.4. Interviews

An individual semi-structured interview using an interview guideline was held with enterprise representatives by video conferencing due to the COVID-19 pandemic. Interviews were pre-tested with 3 SME employers to increase feasibility and comprehensibility of the interview guide. One researcher (FB) conducted and recorded all interviews. One additional researcher (FN) listened back to one interview and gave suggestions for improvement of the interview process. 

The guideline comprised the following parts: the interviewees were asked for an example in which a successful implementation of a preventive health measure had taken place as well as for an example in which an unsuccessful implementation of a preventive measure had taken place. They were then asked to name success factors and barriers that played a role in the respective situation. They were subsequently asked to explain how they would address the failed implementation if they tried again and what strategies they would apply or which resources and other pre-conditions they would need to make it a success. 

## 3. Results

### 3.1. Survey

#### 3.1.1. Respondent Characteristics

In total, 79 respondents from SMEs in the cleaning, care, construction and transport sectors completed the survey. Most were from small enterprises (*n* = 36, 46%) and from the construction sector (*n* = 44, 56%). Details of the enterprises they represented are shown in Table 1. A total of 68 (86%) of respondents indicated physical strain (lifting, carrying, pushing and/or pulling) to be present in their enterprise. A high work demand was reported by 45 respondents (57%). This was followed by computer screen work (*n* = 31, 39%). Aggression and violence and undesirable behavior (e.g., sexual harassment, bullying or discrimination) were least frequently reported to be present.

#### 3.1.2. Use of Preventive Health Measures in SMEs

Measures that were most frequently reported to be implemented were identifying risks with the risk inventory and evaluation tool (RI&E) and drawing up a Plan of Action based on the RI&E (96% and 94% respectively). These are both mandatory by law. On the other hand, health promotion for psychosocial risk prevention (e.g., courses on dealing with stress, time management, relaxation or dealing with aggression and undesirable behavior) and health promotion for musculoskeletal risk prevention (e.g., sports/fitness subscription, lifestyle advice or other sports activities) were less frequently reported to be implemented (38% and 29%, respectively). Furthermore, organizing work differently and adapting the layout of workstations/buildings for better protection against aggression and violence or other undesirable behavior were least frequently reported (23% and 22%, respectively). This is consistent with the low prevalence of the corresponding risk factors. See Table 2 for more detail.

#### 3.1.3. MIDI Determinants of Implementation

A total of 44 (56%) respondents did not indicate that they disagreed or strongly disagreed with any of the MIDI determinants, which could be interpreted as not experiencing any of the determinants to hinder implementation. In total, 16 (20%) respondents indicated that they disagreed or strongly disagreed with only one determinant. Statements on the availability of financial resource and staff were most frequently disagreed or strongly disagreed with (13% and 11% of respondents, respectively). On the other hand, most respondents indicated that it was part of their policy to provide good working conditions and that it is the employer’s task to introduce preventive measures and to provide good working conditions (99% and 95%, respectively). None of the respondents stated that the measures were generally too complicated. Furthermore, all respondents indicated that they regularly inform workers about the available preventive measures. See Figure 1 for a ranking of the determinants. 

### 3.2. Interviews

#### 3.2.1. Respondent Characteristics

In total, 18 enterprise representatives were interviewed; 4 of them were from micro, 6 from small, 4 from medium and 4 from medium–large enterprises. Three of the 18 representatives were working in the cleaning sector, and 5 were in the care, 6 in the construction and 4 in the transport sector. 

#### 3.2.2. Determinants of Implementation

The determinants of implementation that emerged from the analysis of the interviews with enterprise representatives were associated with 10 distinct themes. These were (1) available resources (both finances and staff), (2) complexity of implementation of measures, (3) awareness, (4) knowledge and expertise, (5) availability of time, (6) employer and worker commitment, (7) workers’ openness for measures, (8) communication, (9) workers’ trust and autonomy and (10) integration in organizational policy.

(1)Available Resources: Both Finances and Staff

Some enterprise representatives reported that they lack financial resources for preventive measures, which they attributed to management’s unwillingness to free up resources and competitive pressures due to customers’ unwillingness to accept higher prices for better working conditions as well as ineffective laws and regulations leading to unfair competition with enterprises that make use of “grey areas”. 

“Evolving regulations (on health and safety at work) are always subject to fraudulent interpretation. You have to get rid of that. I have already experienced this with some 20 regulations in recent years. They always create grey areas and you have to get rid of these. You don’t want grey areas; it’s black or white. It is allowed, or it is not allowed, and no ifs or buts.” (cleaning, small-sized enterprise)

In other companies, however, financial resources for preventive measures are an integral part of operational costs, and therefore, this was not perceived as a barrier for implementation. It was also frequently emphasized that measures do not necessarily require additional financial resources. 

“You run into budgetary constraints that limit the amounts you can afford … That becomes especially clear with measures that force you to buy something. Look, other measures don’t cost any money. Changing the duty roster doesn’t cost any money. Slightly increasing the height of a dustbin hardly costs anything, so that’s not the problem.” (care, medium-large-sized enterprise)

The availability of human resources was also considered important. While some enterprises are able to assign one or more staff member to prevention activities, others already have difficulty recruiting qualified workers for the enterprise’s core business. 

“The tricky thing about small companies is that they earn less, have less money to spend and also have less room to burden its personnel with these tasks. The smaller the business, the less attention (prevention) gets. It’s something that someone does in his or her spare time, let’s say, or for an hour a week or early in the morning but not for a whole day because it’s an investment.” (construction, medium-large-sized enterprise)

(2)Complexity of Implementation of Measures

While respondents acknowledged that there are simple measures available to promote and protect workers’ health, they also reported that it means a significant effort to implement some measures, such as the effort associated with identifying and addressing diverse needs and preferences of workers (especially for lifestyle policies), the logistics associated with organizing trainings for workers, the bureaucracy associated with applying for subsidies or the effort associated with identifying and addressing risks, especially when workers are working in diverse and changing environments.

“But the problem with our work is always: we are not a factory. See, if you have a factory, there is the distinction… There you are dealing with a process, and you can always analyze that process, and then you can improve it and also improve ergonomics for the people. And the problem with us in the construction industry is that all situations are pretty much unique.” (construction, small sized-enterprise)

Furthermore, getting workers involved and interested in preventive efforts and changing the culture was seen as a complex task by the representatives. Some additionally reported that there are not always suitable measures available for their specific working circumstances and that not all risks can be prevented. According to them, the innovation of tools is needed to improve working conditions. 

(3)Awareness

Respondents considered workers’ and managers’ awareness with regard to (long-term) health risks, available effective measures and their implementation to be crucial.

First, respondents acknowledged that workers and managers need to be aware of the causal link between certain working conditions or unhealthy behaviors and (long-term) musculoskeletal or mental health problems. Currently, awareness arises mainly when workers have already developed health problems, i.e., when the risk exposure developed into health complaints. However, even then, causality did not always seem to be properly understood, as could be demonstrated in the following quote by an enterprise owner who had worked as a cleaner in the past: 

“Now I’m also not like… that my symptoms are due to my work. Sure, I did a lot of mechanical moves with my arms and my shoulders, my legs, my feet. Walked a lot, climbed stairs a lot. But is that related to the fact that I worked with heavy materials, or is it just my natural predisposition? I don’t know. I do believe in wear and tear, but my colleague who is the same age doesn’t have it at all, and she has lugged just as much. So where is the difference? You don’t know. So it’s hard to demonstrate.” (cleaning, small-sized enterprise)

Awareness of risks that workers encounter in their day-to-day jobs was also cited as critical to take appropriate preventive action. However, respondents reported that not all psychosocial and musculoskeletal risks are systematically identified. Risk assessment is often based on “common sense” and depends on workers letting managers know when they encounter problems: 

“Well, if you want to sexually intimidate a receptionist, the whole company hears about it. So, it doesn’t happen. And I don’t get any complaints from them either that they experienced anything annoying. Because we can always discuss it if something is wrong. So, I’m not in a big hurry about it.” (construction, medium-sized enterprise)

However, this openness was reported to potentially be hindered by a lack of trust and communication. This was mentioned as particularly problematic in the case of preventing mental health disorders, which are not only poorly understood but can also be stigmatized by co-workers or employers.

(4)Knowledge and Expertise

It was mentioned that it is important that workers are well informed about newly introduced measures and how to apply them. To verify that this is the case, managers need to have insight into how measures are actually being applied in practice or why they are not being applied. However, this can be difficult, especially when workers are working at different locations.

“But there is always another dimension, and that is what does the employee herself do once she has received these instructions? Will she continue to do what she has been told, in order to make things better for herself? And because we don’t always have a good view on that because we don’t see these ladies working every day, it’s quite difficult. … Because you can’t correct someone. If you would see them every day and observe that they work in certain ways, or in the wrong way, then you can correct someone, help them or explain it again or make them go to a training course again.” (cleaning, medium-sized enterprise)

External professionals, e.g., occupational health services or physical therapists, can drive implementation by coaching, advising or assisting managers with implementation or by providing interventions directly to workers. However, some reported that these parties currently focus too much on reintegration rather than prevention and do not know enough about day-to-day working conditions to provide adequate support. Respondents also reported that they lack knowledge and expertise to assess the quality of commercial service providers, such as providers of communication skill training courses, and to compare different providers based on their effectiveness (and not just on their price).

(5)Availability of Time

The availability of time plays a role for both employers and workers. Paradoxically, according to respondents, it is not always the actual availability of time that determines the use of measures, but rather a perceived availability of time by workers. Some interviewees thus attributed a latent feeling of lack of time among workers to culturally induced diligence and team dynamics rather than to an actual lack of time. 

“What you notice then is that the speed of work is actually somewhat reduced by the use of these tools. So people are very quick to leave the tools behind and continue to work in the old-fashioned way, … because that is faster, while nobody is asking them to.” (cleaning, small-sized enterprise)

(6)Employer and Worker Commitment

The commitment and support of the employer was identified to depend largely on the extent to which they are convinced that the prevention measures will actually lead to a competitive advantage, but also on their personal sense of responsibility toward workers. The responsibility towards workers seemed to be particularly strong in small companies where (owner-) managers and employees meet regularly or even work side by side:

“…for me, the enterprise comes first, but, in addition, the interests of the personnel do not come second but rather at place one and a half.” (cleaning small-sized enterprise)

While most representatives agreed on their responsibility for providing working conditions that protect and promote workers’ health, they were less unanimous about their responsibility for living conditions and behaviors outside the work context. 

Instruments to strengthen the commitment of employers (e.g., inspection and certification) were sometimes recognized as effective: 

“Then, the (certification agency) sends out this signal: hey, we must, and that sounds a bit annoying because of this ‘we must’, but ‘must’ often works better than saying: hey, on a voluntary basis. … And I think that most SME’s don’t actively look for that kind of information for the fun of it, because you’re busy with other things too. … So, you’ll have to keep that compulsory character, I think.” (construction, medium-sized enterprise)

However, some enterprise representatives feared that these incentive schemes would become an end in themselves, undermining intrinsic motivation. 

On the other hand, it was emphasized that the success of prevention efforts depends on workers helping to shape the measures and the implementation process so that they meet their needs and expectations and so that they ultimately use the available measures. While some enterprises struggle with a lack of worker interest, others reported that workers were enthusiastic about using an intervention or embracing an opportunity to help shape the implementation process:

“…there are enough people who like that. … Well, of course it’s nice to engage with new technology, to engage with new tools, to be allowed to participate in decision making, to feel responsible, to be involved in general. Well, you might rather ask who wouldn’t want that.” (care, medium-large sized enterprise)

(7)Workers Openness for Measures

While some enterprise representatives, especially in the care sector, reported workers who have internalized the importance of healthy living and working and a company culture where workers are open about their health-related needs and encourage and support each other to live and work healthily, this was not yet the case in other companies. Particularly in teams with predominantly male workers, e.g., in the construction industry, workers were reported not to dare to talk about health-related issues and needs and act according to the motto “Don’t complain, get on with it”. Furthermore, older workers who have already developed unhealthy routines may negatively impact behaviors of younger workers who would be more open to change. 

Openness to make an effort to change behavior was also reported to be essentially influenced by the expected effectiveness of measures (in the long term). If workers understand the reasons for measures and their impact, they are more motivated to help shape and use them. However, this was seen as difficult to realize, as many measures only have effects in the long-term.

“That they say, hey, look, we’re not providing all kinds of tools for nothing. It’s not meant to bother you. It’s to make sure that later … you’ll be riding your bike with your grandchildren when you’re 75, and they won’t have to push you around in a wheelchair.” (construction, medium-large-sized)

(8)Communication

Respondents emphasized the importance of open and target group-oriented communication about (long-term) risks and needs, available measures and their appropriate implementation and application. While, in some companies, health-related needs are openly communicated, adequate risk reporting systems have been established, and workers are regularly informed and encouraged to share their experiences, others reported that they have difficulties in communicating prevention-related information (e.g., about risks and available measures and their application) to workers in an accessible way.

“…this group does not want to read through texts. They want something brief but powerful. For example, a short film about how do you lift properly, when do you lift in a harmful manner? And just some reminders and some fresh-up stuff. Sometimes I’m also on the lookout: how am I going to inform this group?” (cleaning, small-sized enterprise)

Representatives also reported that they themselves need to be approached in an appropriate way by external stakeholders involved in the implementation process, e.g., sector organizations, labor inspections, health and safety services, legislators and knowledge centers. Some information was perceived to be too complex, too exuberant and not tailored to the specific needs of small businesses. 

(9)Workers’ Trust and Autonomy

Respondents reported that employers’ attempts to gain insights into workers’ health needs and to offer them interventions may be perceived as an invasion of their privacy. Furthermore, such efforts may cause mistrust if workers do not believe that the employer is actually interested in their health: 

“Well, he’s doing it to his own advantage, they think.” (cleaning, small-sized enterprise)

Enterprise representatives therefore reported refraining from acting to maintain good relationships and let workers have their autonomy rather than (being under the suspicion of) patronizing them. A good relationship between employers and workers, based on trust, was reported to be the driver for implementation. It makes it easier for workers to communicate their needs and thus easier for employers to identify needs.

“And, well, the most important thing is that there is a good atmosphere, openness. People must have the confidence that they can discuss anything here, whatever they want to share with us. And of course, I can’t possibly ask everyone all the time. So that sense of responsibility is really necessary and what they want themselves as well. They can ask a lot of me, but I also ask them to ring the bell if there is something wrong. I can’t always spot it.” (care, small-sized enterprise)

(10)Integration in Organizational Policy

Another determinant of implementation described by the representatives was the extent to which the implementation process is effectively designed. It was suggested that one or a group of workers who take responsibility and liability for prevention efforts and who can be easily approached by workers is important. These persons need to be trained and given temporal resources for prevention work. The lack of such an officially appointed coordinating body can lead to prevention being one of many tasks falling under the responsibility of one person, which is often neglected because of other priorities. 

Paying attention to the availability of preventive measures on a regular basis and over a long period of time to raise workers awareness and to de-taboo the topic of health complaints was another aspect being identified. While some enterprise representatives acknowledged regular scheduled meetings on these topics to be useful, others found this unnecessary, as they see and talk to workers every day on these aspects. This was especially true for micro enterprises: 

“They are not my friends, but I have a lot of contact with them. Everyone can walk in freely too. … So come on, throw it in the group. People need to get rid of meeting cultures to then have to discuss things. Toolbox meetings, yes, because then we’re going to discuss how the work can be done differently, and then nobody opens their mouth because they don’t dare to.” (cleaning, small-sized enterprise) 

## 4. Discussion

The aim of this study was to provide an overview of the current use of measures to prevent musculoskeletal and mental health disorders in SMEs in the Netherlands and to identify and examine determinants of the implementation of those measures from the perspective of the employer or other enterprise representatives. 

SME representatives from the cleaning, care, construction and transport sectors in the Netherlands, whose perspectives have been shed light on in this study, unanimously agree on their responsibility to prevent musculoskeletal and mental health disorders at work. However, their sense of responsibility seems to be mainly limited to the provision of those preventive health measures that focus on improving working conditions, as well as those that are required by law. So-called “lifestyle interventions” aimed at encouraging workers to adopt healthier lifestyles, such as sports/fitness subscriptions, lifestyle counselling or stress-management courses, are less frequently adopted.

Determinants of implementation according to enterprise representatives were associated with 10 distinct themes. These were (1) available resources (both finances and staff) (2) complexity of implementation of measures, (3) awareness, (4) knowledge and expertise, (5) availability of time, (6) employer and worker commitment, (7) workers’ openness for measures, (8) communication, (9) workers’ trust and autonomy and (10) integration in organizational policy. The determinants varied widely. This is attributable to the heterogeneity of contexts (enterprise sizes and sectors), targeted risks (psychosocial vs. musculoskeletal) and the scope of preventive health measures whose implementation was researched in this study. The following elaboration will focus on three aspects that stood out. Firstly, the fact that, according to enterprise representatives, determinants mainly lie within the sphere of influence of the workers; secondly, the relevance of the availability of resources and, thirdly, the relevance of good communication.

For the implementation of preventive health measures to be successful, according to the interviewees in this study, workers need to have awareness, knowledge and expertise with regard to risks and available preventive measures; take sufficient time to apply them; support prevention efforts; be engaged and participate in them; be open to change; communicate their needs clearly and have trust in the goodwill of their employer. Enterprise representatives thus seem to shift the responsibility for successful implementation onto workers. This tendency towards the “individualization” of prevention has also been described in the context of the SESAME project, which researched the implementation of occupational health and safety measures in micro and small enterprises with up to 50 workers [13]. However, it may not always be realistic for workers to meet all the above requirements. Reasons that may hold workers back from taking care of their health during and outside work may be a low health literacy of workers in the sectors studied [14], a precarious labor market position and employment contract putting them under pressure to trade off their health with better work outcomes [15] and the fear of not being taken seriously or stigmatized, especially when it comes to psychosocial risks [16]. Hasle et al. [13] stressed that it is, among other things, because of these factors that it is by law the responsibility of the employer to make sure that preventive measures are not only adopted, but also successfully implemented. However, they also found that this responsibility is often assumed by workers without complaint. The development of implementation strategies targeting workers can therefore only be part of a holistic strategy to improve the implementation of preventive health measures. 

One factor most frequently mentioned in the survey as not being met is the availability of resources in the form of finances and staff to implement preventive health measures. The relevance of resource availability was also found in previous studies [17,18,19]. By respondents in our interviews, this has mainly been attributed to competitive pressures. The literature suggests that these have increased in recent years due to, among other reasons, what has been labelled “workplace fissuring” [20]. Larger organizations, faced by pressure to improve financial performance for private and public investors, transfer responsibility for (parts of) their operations to SMEs or even to the workers themselves (through contract labor and the gig economy). Because of the greater bargaining power of these larger organizations, SMEs depend on their goodwill and willingness to pay for better working conditions. On the other hand, they have to compete with the (pseudo) self-employed. This can put them in an economically precarious situation, leaving little room to invest in prevention. The promotion of low cost measures and inventive opportunities for funding measures such as group purchasing of measures via employer organizations, which has been suggested by Harris et al. [18]. should therefore be explored in future research. 

Our study also identified the importance of communication for successfully implementing preventive measures in SMEs, which confirms previous research findings [21]. Communication plays a role at different levels. External actors (governments, regulators, sector associations and other intermediaries) need to use communication strategies that are specific to SME preferences, and employers need to communicate effectively with staff. Conversely, staff must effectively communicate their needs to employers. Previous research suggests that communication strategies benefit from breaking down the term ‘small business’ into smaller, more homogeneous categories that can subsequently be targeted specifically [21]. Champoux et al. [22] analyzed the most characteristic prevention measures and practices in small companies and found four different profiles of enterprises that could potentially be used as a basis to develop targeted communication strategies. These profiles differ in the extent to which enterprises actively work on prevention, the extent to which enterprise representatives are informed about risks and available measures, whether the employer acts “traditionally” and assumes all responsibility for prevention or they involve employees in prevention efforts and whether they have formal structures and are involved in networks where they can learn from others about prevention [22]. The four profiles are thus “the inactive and uninformed”, “the inactive, traditional, unstructured”, “the active, participative, unstructured” and the group of enterprises that are active, participative, structured and integrated in networks. 

## 5. Strengths and Limitations 

Low participation rates are common in research with this target group, and we have not been spared this problem either. In order to increase the response rate of the survey, the target group was approached via different channels, so we have no insight into exact participation rates due to the unknown denominator. Consequently, this report presents results of a selective sample of all firms in these sectors. The fact that interviewees were recruited via the preceding survey also adds to the risk of selection bias. This led to a more positive picture than would have been found in a representative sample of SMEs. For example a recent survey among Dutch enterprises showed that only 52% had performed a RI&E and that 78% had drawn up a plan of action based on this RI&E (as opposed to 96% and 94% of our survey respondents) [9].

Some reports also suggest that the perception of enterprise representatives may be distorted and more positive than the actual situation. As a consequence, further research is being conducted by this research team to explore the perception of other stakeholders, such as workers, members of control authorities or occupational physicians. 

Despite these limitations, the combination of the survey as well as interview data can be mentioned as an advantage. The findings of this exploratory study are thus practically useful for stakeholders in the field of health prevention in SME as well as for researchers to improve strategies for implementing preventive health measure in SMEs. 

## 6. Conclusions

This study has provided more insight into the determinants of the implementation of preventive health measures in SMEs. The findings can serve as a support for developing strategies for implementing preventive health measures. The heterogeneity of the results across different sectors, enterprise sizes, types of interventions and health risks suggests that there is no one-size-fits-all solution but that implementation strategies need to be tailored to the specific situation of the enterprise and take different determinants into account. Next to the views of enterprise representatives, the workers’ views and that of other stakeholders should also be explored in order to get a more complete picture of the implementation problem.

## Figures and Tables

**Figure 1 ijerph-19-03904-f001:**
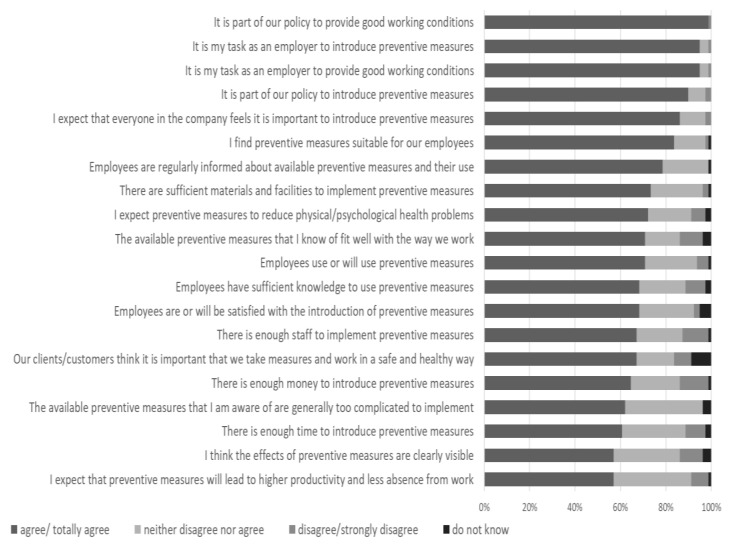
MIDI determinants of implementation (*n* = 79).

**Table 1 ijerph-19-03904-t001:** Respondent characteristics.

Characteristic	Frequency (*n* = 79)
Enterprise size	
Micro (workers ≤ 9)	12 (15%)
Small (10 ≤ workers ≤ 49)	36 (46%)
Medium (50 ≤ workers ≤ 99)	13 (16%)
Medium–large (100 ≤ workers ≤ 250)	18 (23%)
Sector	
Cleaning sector	8 (10%)
Care	14 (18%)
Construction	44 (56%)
Transport sector	13 (16%)
Self-reported prevalence of risk factors (several answers possible)	
Physical strain (lifting, carrying, pushing and/or pulling)	68 (86%)
High work demands	45 (57%)
Computer screen work	31 (39%)
Long-lasting (uncomfortable) working positions	15 (19%)
Irregular working hours	13 (16%)
Repetitive work	10 (13%)
Emotionally demanding work	8 (10%)
Aggression and violence	3 (4%)
Undesirable behavior (e.g., sexual harassment, bullying or discrimination)	2 (3%)

**Table 2 ijerph-19-03904-t002:** Use of preventive health measures.

Name of the Preventive Health Measure	Frequency (*n* = 79)
Self-reported measures for musculoskeletal risk prevention (several answers possible)	
Providing aids to make the work less physically demanding	73 (92%)
Rearranging the workplace to improve posture	70 (89%)
Organizing work differently (e.g., more variety in the work or better allocation of tasks)	66 (84%)
Instruction, training and supervision, e.g., about correct lifting techniques or posture	64 (81%)
Personal protective equipment (e.g., knee pads or protective clothing)	52 (66%)
Health promotion (e.g., sports/fitness subscription, lifestyle advice or other sports activities)	23 (29%)
Self-reported measures for psychosocial risk prevention (several answers possible)	
Offering training, courses or guidance for personal development	70 (89%)
Making better agreements on the division of work	68 (86%)
Social activities to promote social coherence	65 (82%)
Adapting tasks to better suit workers’ competences and skills	64 (81%)
Deploying extra workers/helping out when it is busy	64 (81%)
Making better agreements about working hours	58 (73%)
Appointing a confidential advisor and help for workers who are bothered by undesirable behavior	54 (68%)
Agreements and instructions about undesirable behavior	52 (66%)
Health promotion (e.g., courses on dealing with stress, time management, relaxation or dealing with aggression and undesirable behavior)	30 (38%)
Organizing work differently for better protection against aggression and violence or other undesirable behavior	18 (23%)
Adapting the layout of workstations/buildings for better protection against aggression and violence or other undesirable behavior	17 (22%)
Self-reported general preventive measures (several answers possible)	
Identifying risks with the risk inventory and evaluation tool (RI&E)	76 (96%)
Drawing up a Plan of Action based on the RI&E	74 (94%)
Availability of an Occupational Health Service (OHS) or occupational physician for advice on improving working conditions/health	71 (90%)
Offering workers a health check or employability check	60 (76%)
Appointing an employee who is (also) specifically concerned with improving working conditions (prevention officer)	59 (75%)
Employee satisfaction surveys or work pressure surveys	37 (47%)

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
