# Peer review of "The Implementation of Preventive Health Measures in Small- and Medium-Sized Enterprises—A Combined Quantitative/Qualitative Study of Its Determinants from the Perspective of Enterprise Representatives"

_ijerph, 2022, doi:10.3390/ijerph19073904_

Round 1
Reviewer 1 Report
Manuscript ID: ijerph 1608161
The implementation of preventive health measures in small and medium- sized enterprises – A combined quantitative/qualitative study of its determinants from the perspective of enterprise representatives.
The paper explores preventive health measures in small and medium enterprises using a mixed method approach. The paper is certainly relevant and provides some insights, but it is incomplete. The main reason is that there is no assessment of whether the measures adopted are successful. This can only be said looking at the workers’ outcomes (i.e. mental health etc). The implementations of the measures would not necessarily translate into efficacy of such measures. The only way to assess its success is to look at workers' outcomes. I
Main comments:
- This is certainly a nice paper, but one issue is the size of the sample, especially when using quantitative methods. The author(s) do discuss a bit the sample issue but not the implications for quantitative methods.
- Interviews were conducted during the second stage of Covid 19. It is well known that, for example, mental health of workers especially in some sectors as those of the analysis got worse during this time. It would perhaps worth saying something about this. Have the requirements for SME increased?
- I wonder whether the measures implemented in the workplaces, like courses were compulsory for workers? Section 3 discusses the awareness of workers but it does not mention whether the trainings are compulsory, this could have some implications for the efficacy of the measures.
- While it is clear that the aim of the study is to analyse the measures from an employer point of view, I am not sure how this can be separated from workers. The study does not say anything about the consequences of taking such measures. Specifically the study does not say if the measures have been successful, and will only able to say something if could interview some workers.
- It seems to me that this study is mainly qualitative. The quantitative elements reported are just a couple of frequency tables with no statistical significance. So I am unsure whether the study can be defined quantitative. Clearly the qualitative element prevails.
Minor comments:
- Abstract, line 16 this sentence could be improved as there is a repetition ‘the implementation of preventive measures to prevent…’
- Abstract: I am not sure I would list all 10 determinants in the abstract.
Author Response
We thank the reviewer for taking the time to read our manuscript and provide feedback and suggestions for it. Below you will find a pointwise response to each of the reviewers comments. We marked the changes made in the manuscript via “track changes”.
Reviewer 1:
The implementation of preventive health measures in small and medium- sized enterprises – A combined quantitative/qualitative study of its determinants from the perspective of enterprise representatives.
The paper explores preventive health measures in small and medium enterprises using a mixed method approach. The paper is certainly relevant and provides some insights, but it is incomplete. The main reason is that there is no assessment of whether the measures adopted are successful. This can only be said looking at the workers’ outcomes (i.e. mental health etc). The implementations of the measures would not necessarily translate into efficacy of such measures. The only way to assess its success is to look at workers' outcomes.
- The aim of this study is to explore the determinants of the implementation of preventive health measures in SMEs. The efficacy of such measures is indeed an interesting research question to address, but is beyond the scope of this manuscript.
Main comments:
This is certainly a nice paper, but one issue is the size of the sample, especially when using quantitative methods. The author(s) do discuss a bit the sample issue but not the implications for quantitative methods.
- We are indeed aware of the potential selectiveness of the sample. The implications of this issue are discussed in lines 543-550: ”Low participation rates are common in research with this target group, and we have not been spared this problem either. In order to increase the response rate of the survey, the target group was approached via different channels, so we have no insight into exact participation rates due to the unknown denominator. Consequently, this paper may present results of a selective sample of all firms in these sectors. The fact that interviewees were recruited via the preceding survey also adds to the risk of selection bias. This may have led to a more positive picture than would have been found in the general population of SME representatives.”
Interviews were conducted during the second stage of Covid 19. It is well known that, for example, mental health of workers especially in some sectors as those of the analysis got worse during this time. It would perhaps worth saying something about this. Have the requirements for SME increased?
- The reviewer has a good point in that the mental health of workers in some sectors may have worsened, this has been studied in Hannemann et al. (Hannemann, J., Abdalrahman, A., Erim, Y., Morawa, E., Jerg-Bretzke, L., Beschoner, P., & Albus, C. (2022). The impact of the COVID-19 pandemic on the mental health of medical staff considering the interplay of pandemic burden and psychosocial resources—A rapid systematic review. PloS one, 17(2), e0264290). However, our study explored determinants of the implementation of preventive (mental) health measures in SMEs, but did not aim to examine the mental health prevalence of workers. Although we have no data about the determinants of the implementation of preventive health measures during the pandemic, we do not expect these to be significantly affected by the pandemic.
I wonder whether the measures implemented in the workplaces, like courses were compulsory for workers? Section 3 discusses the awareness of workers but it does not mention whether the trainings are compulsory, this could have some implications for the efficacy of the measures.
- Making preventive measures, such as a training, compulsory is indeed one strategy to improve the implementation of preventive measures. The objective of our study was to explore the determinants of the implementation of preventive health measures. The efficacy of the measures was not studied.
While it is clear that the aim of the study is to analyse the measures from an employer point of view, I am not sure how this can be separated from workers. The study does not say anything about the consequences of taking such measures. Specifically the study does not say if the measures have been successful, and will only able to say something if could interview some workers.
- We agree with the reviewer that the employee perspective is relevant and therefore we are working on another study that aims to shed light on their perspective on the determinants of the implementation of preventive health measures. As a separate and different data collection among workers in SMEs was needed, the study of the employers’ and workers’ perspective could not be combined in one manuscript.
It seems to me that this study is mainly qualitative. The quantitative elements reported are just a couple of frequency tables with no statistical significance. So I am unsure whether the study can be defined quantitative. Clearly the qualitative element prevails.
- The reviewer is right that the qualitative analysis prevails in this study, but it is complemented by the descriptive results of the quantitative (survey) data. Therefore, we consider it a combined quantitative-qualitative study (and not a qualitative study only) where both quantitative and qualitative data contribute to the study aim.
Minor comments:
Abstract, line 16 this sentence could be improved as there is a repetition ‘the implementation of preventive measures to prevent…’
- We revised the sentence according to this comment: “The aim of this study is to investigate determinants for the implementation of measures to prevent musculoskeletal and mental health disorders from the perspective of enterprise representatives in Dutch SMEs.”
Abstract: I am not sure I would list all 10 determinants in the abstract.
- We agree that this is indeed a long list of determinants but it is in our view relevant for the abstract since it provides the reader an immediate overview of the main results of the manuscript.
Reviewer 2 Report
Indeed, the work environment is an ideal place for the implementation of public policies. However, in the field of SMEs it is more difficult since they have another type of work structure. Occupational diseases such as those related to musculoskeletal and psychosocial risks are relevant and therefore it would be interesting to learn about these issues in the field of SMEs.
Online surveys are not adequate given that they are accompanied by the biases typical of a cross-sectional study. In addition, biases typical of online studies are added. Although the researchers affirm that they will do some interviews, but this is not necessary, they will be able to control the biases indicated above. It would be advisable to indicate how these possible biases will be controlled.
What would be the universe of the study to be able to determine what % of response corresponds to a n=79 that are the workers who answered. Therefore, the results could not be extrapolated to the rest of the group. I recommend placing the universe to understand what the response rate was.
The researchers suggest changes related to work organization, infrastructure, training, and behavioral changes. My question is whether these actions have been discussed with the SMEs or are only changes suggested based on the results of the study possible?
A common problem is recognizing that some musculoskeletal conditions are difficult to recognize as work-related illnesses. Above all, in cases where there are individual risk factors that are important to address. Lifestyles can be promoted, but ultimately it is individuals who must take responsibility for them. Likewise, it should be considered that workers spend a large part of their time in the workplace, so health promotion should be insisted on without forgetting the rest of the family. For example. In the workplace you can encourage healthy eating and have a canteen that offers healthy eating, but at home the diet can be different as it follows the family rhythm. The question would be, if the family is involved in promoting healthy lifestyles or is it just limiting itself to the scope of the company.
A final concern is whether an Ethics Committee approved this project and whether informed consent was applied to the participants.
Author Response
We thank the reviewer for taking the time to read our manuscript and provide feedback and suggestions for it. Below you will find a pointwise response to each of the reviewers comments. We marked the changes made in the manuscript via “track changes”.
Indeed, the work environment is an ideal place for the implementation of public policies. However, in the field of SMEs it is more difficult since they have another type of work structure. Occupational diseases such as those related to musculoskeletal and psychosocial risks are relevant and therefore it would be interesting to learn about these issues in the field of SMEs.
Online surveys are not adequate given that they are accompanied by the biases typical of a cross-sectional study. In addition, biases typical of online studies are added. Although the researchers affirm that they will do some interviews, but this is not necessary, they will be able to control the biases indicated above. It would be advisable to indicate how these possible biases will be controlled.
- The frequency tables merely give insights into a trend on which of the preventive measures have been implemented in the surveyed enterprises. They do not provide information on the implementation rate in the general SME population. This is discussed in lines 543-552: “Low participation rates are common in research with this target group, and we have not been spared this problem either. In order to increase the response rate of the survey, the target group was approached via different channels, so we have no insight into exact participation rates due to the unknown denominator. Consequently, this report presents results of a selective sample of all firms in these sectors. The fact that interviewees were recruited via the preceding survey also adds to the risk of selection bias. This led to a more positive picture than would have been found in a representative sample of SMEs. For example a recent survey among Dutch enterprises showed that only 52% had performed a RI&E and 78% had drawn up a plan of action based on this RI&E (as opposed to 96% and 94% of our survey respondents) [9].”
What would be the universe of the study to be able to determine what % of response corresponds to a n=79 that are the workers who answered. Therefore, the results could not be extrapolated to the rest of the group. I recommend placing the universe to understand what the response rate was.
- We are not sure what is meant by ‘universe’, but due to our broad recruitment strategy, it is unfortunately not possible to determine the number of enterprises that have received the survey. This is elaborated on in lines 101-115: “In order to reach enterprise representatives three different strategies were applied. First contact persons referred to us by existing contacts within sector organizations in the cleaning, care, transport and construction sectors were informed about the study and asked to distribute the survey among their SME members. In doing so, the contact persons were provided by the research team with a flyer with brief information about the study, a recruiting text to post on their website, social media or the newsletter and the link to the survey. Each contact person used their own existing and usual communication channels to approach employers. These included newsletters, social media sites and face-to-face contact. Second, several other organizations such as local employer associations, the Netherlands Society of Occupational Medicine and a network for HR professionals, were approached and asked to distribute the recruitment material. Third, enterprise representatives participating in the survey were asked to send the survey to other enterprise representatives. The data collection took place from February 2021 until July 2021. In total 79 respondents in our target group completed the survey. Although we cannot specify the response rate because of this broad recruitment approach, we assume it is very low.”
The researchers suggest changes related to work organization, infrastructure, training, and behavioral changes. My question is whether these actions have been discussed with the SMEs or are only changes suggested based on the results of the study possible?
- Interviewees where informed about the type of measures they should think of when answering to the interview questions about the determinants of the implementation of preventive health measures. We gave them the following definition, which can also be found in lines 93-98: “Definition of preventive health measures: In this research a preventive health measure is defined as an attempt to improve musculoskeletal and mental health conditions in workplaces by means of targeted activities and initiatives. Such activities include amongst others changes in work organization and working conditions, engineering activities for the modification or installation of plant and equipment, training and behavioral changes.”
A common problem is recognizing that some musculoskeletal conditions are difficult to recognize as work-related illnesses. Above all, in cases where there are individual risk factors that are important to address. Lifestyles can be promoted, but ultimately it is individuals who must take responsibility for them. Likewise, it should be considered that workers spend a large part of their time in the workplace, so health promotion should be insisted on without forgetting the rest of the family. For example. In the workplace you can encourage healthy eating and have a canteen that offers healthy eating, but at home the diet can be different as it follows the family rhythm. The question would be, if the family is involved in promoting healthy lifestyles or is it just limiting itself to the scope of the company.
- The scope of this study is restricted to the enterprise context.
A final concern is whether an Ethics Committee approved this project and whether informed consent
was applied to the participants.
- As the study did not meet the criteria of the Medical Research Involving Human Subjects Act, This was exempted for further approval by an ethical research committee. Please view lines 584-590.
- Also we added a paragraph on this in the methods section lines 125-130: “The Center for Clinical Expertise of the Dutch National Institute of Public Health and the Environment assessed the research proposal and classified the study as exempt from ethical review as it did not meet the criteria of the Medical Research Involving Human Subjects Acts. The center approved the study protocol. The study was performed in accordance with guidelines of good clinical practice and ethical principles as stated in the Declaration of Helsinki. Informed consent was obtained from all participants.”
Round 2
Reviewer 1 Report
Dear Authors
many thanks for responding to my comments, that I hope have been useful.
I did enjoy reading your paper, and despite limitations as previously indicated and recognised in the paper, this remains an interesting contribution.